# The Automatic Detection of Cognition Using EEG and Facial Expressions

**DOI:** 10.3390/s20123516

**Published:** 2020-06-21

**Authors:** Mohamed El Kerdawy, Mohamed El Halaby, Afnan Hassan, Mohamed Maher, Hatem Fayed, Doaa Shawky, Ashraf Badawi

**Affiliations:** 1Center for Learning Technologies, University of Science and Technology, Zewail City, Giza 12578, Egypt; melkerdawy@zewailcity.edu.eg (M.E.K.); afhassan@zewailcity.edu.eg (A.H.); s-moa.maher@zewailcity.edu.eg (M.M.); abadawi@zewailcity.edu.eg (A.B.); 2Mathematics Department, Faculty of Science, Cairo University, Giza 12613, Egypt; halaby@sci.cu.edu.eg; 3Mathematics Program, University of Science and Technology, Zewail City, Giza 12578, Egypt; hfayed@zewailcity.edu.eg; 4Engineering Mathematics Department, Faculty of Engineering, Cairo University, Giza 12613, Egypt

**Keywords:** cognitive skills measurement, electroencephalography, facial expressions, deep and shallow learning

## Abstract

Detecting cognitive profiles is critical to efficient adaptive learning systems that automatically adjust the content delivered depending on the learner’s cognitive states and skills. This study explores electroencephalography (EEG) and facial expressions as physiological monitoring tools to build models that detect two cognitive states, namely, engagement and instantaneous attention, and three cognitive skills, namely, focused attention, planning, and shifting. First, while wearing a 14-channel EEG Headset and being videotaped, data has been collected from 127 subjects taking two scientifically validated cognitive assessments. Second, labeling was performed based on the scores obtained from the used tools. Third, different shallow and deep models were experimented in the two modalities of EEG and facial expressions. Finally, the best performing models for the analyzed states are determined. According to the used performance measure, which is the f-beta score with beta = 2, the best obtained results for engagement, instantaneous attention, and focused attention are EEG-based models with 0.86, 0.82, and 0.63 scores, respectively. As for planning and shifting, the best performing models are facial expressions-based models with 0.78 and 0.81, respectively. The obtained results show that EEG and facial expressions contain important and different cues and features about the analyzed cognitive states, and hence, can be used to automatically and non-intrusively detect them.

## 1. Introduction

Detecting cognitive load during learning is a critical challenge for the progress of educational technologies, especially in computer-supported learning platforms [1]. The question to be answered is whether designed systems and content delivery impose too many cognitive demands on a learner. What makes the problem even harder is that not only cognitive load varies from one learner to another, but it also varies for the same learner and for the same material depending on other factors that might affect how they perceive the content delivered. For instance, the affective states of learners and how motivated they are have an influence on how they effectively learn [2,3,4]. Thus, a personalized learning system needs to extend and improve cognition, affection, and metacognition of learners based on the collected physiological data [5]. Cognition refers to learners’ ability to perceive, store, process, and retrieve information. Affection is the set of academic emotions such as frustration, engagement, boredom, etc. As for metacognition, it refers to self-regulation such as attention and strategy selection.

Many tutoring systems were designed to manage cognitive load based on behavioral and interface data. However, only a few studies have explored the use of physiological monitoring tools to make more nuanced pedagogical decisions. Those tools include any device that can monitor and sometimes record bio-signals that are thought to be affected by cognition and emotions, and hence, they affect the learning process. Examples of such tools include depth-sensing cameras, electroencephalography (EEG), electro-dermal activity (EDA) sensors, eye and gaze tracking, posture and seat pressure detectors, and many others [6].

In this study, we focus on two input modalities, namely, EEG and facial expressions. The main research question to be answered is how extracted information from these two modalities would reflect and model cognition, and hence, information processing abilities of learners.

EEG has been utilized before to monitor cognitive states and brain activity. It is widely used to measure the changes in electrical voltage on the scalp, induced by cortical activity [7]. Various EEG-based studies have been proposed in various applications, e.g., in detecting driver drowsiness [8,9,10], in human computer interaction [11], and in detecting engagement in learning settings [12,13,14]. In addition, the affective aspects of learning are very important to ensure an effective learning and engagement [15,16]. For example, adapting to students’ uncertainty, confusion, or lack of motivation is a proven technique to significantly enhance the efficiency of teaching. Hence, there is a growing interest in efficient assessments of the psychological status of learners, such as physiology sensors of emotions and self-reporting methods. Moreover, facial expressions as a method of psychological assessment is especially the subject of many studies because of the relative ease of collecting them in a learning environment, and the cumulative knowledge of their analysis methods from other domains of application [17].

The approach utilized in this study consists of the following main steps. First, the subject takes two scientifically-validated tests that measure different cognitive skills and states of interest. Simultaneously, the subjects’ EEG signals and facial expressions are collected using a wireless headset and video recording, respectively. Second, features are extracted from both the EEG signals and video records. Third, different machine and deep learning approaches are utilized to build models of cognitive states and skills of interest using the extracted features.

The next sections present the following. In Section 2, a thorough investigation of related literature is presented. In Section 3, the materials and methods utilized in data collection, artifact removal, and features extraction are discussed. In Section 4, the results of the analysis with special focus on the best performing models are presented. Finally, in Selection 5, a summary of the conclusions of the analysis and future work are presented.

## 2. Related Work

The literature contains a variety of studies that affirm the EEG signals as a primary input to investigate cognitive states and skills in different application domains [12]. The range of cognitive states and skills investigated in the literature using EEG signals includes alertness, impairment, drowsiness, memory, attention, memory, and workload level [13,14,15,16,18,19]. The main approach utilized by most of these studies is to employ machine learning techniques to build a modality for the cognitive skill or state investigated using features extracted from the EEG signal. The extracted features usually include time domain features and frequency domain features.

EEG was employed to determine the cognitive state during the learning of the subject; in addition, an intelligent tutoring system (ITS) was used [20]. This ITS designed their educational strategies dynamically, where a certain amount of cognitive load was assigned to each participant. EEG data was collected from the students wearing a hat-like headset while learning from ITS. Models EEG spectral features were used in designing models that used partial least squares regression to differentiate between the cognitive load of the training set; high and low ones. It also used linear mixed effects regressions to relate the questions performance to certain tasks.

EEG was also used in [21] for the encoding of cognitive load in multimedia learning tasks; multimedia animations based on plant anatomy. In this study a 128-channel device was used at a sampling frequency of 250 Hz in addition to recording eye blinks and heart rate to help the preprocessing of EEG signals. After the multimedia learning task, a memory recall test was given to the subjects where three spectral features were extracted using four distinct frequency bands. The connectivity patterns among the selected channels of the brain was determined using partial directed coherence (PDC). These three spectral features were classified using three different classifiers; Naïve Bayes, SVM with linear kernel, and SVM with radial basis function kernel, and the best accuracy was obtained using alpha waves.

In [22], working memory using different tasks including verbal n-back, spatial n-back, and the multi-attribute task battery [23] was measured. In this study, a 30-channel EEG was collected in a room electrically and magnetically-shielded and the sum of power spectral density (PSD) was calculated in seven frequency bands using Burg’s method. A total of seven features for each of the 30 channels were used. Good results were obtained when a support vector machine (SVM) with linear kernel function was used. In [24], an approach for the characterization of attention using EEG is presented. A sample of 450 data points generated from 10 subjects was used. The used features included 25 features, which included the power density integral, the peak power frequency (center frequency), and the max power density (maxpower). Different feature selection and shallow learning models were used in experiments. In the end, the best classification result was obtained using CFS+KNN, with an accuracy of 80–84% (+/− 3%). The study also asserted that the beta bands contributed to the most important features to the model. Another study that investigated how attention can be reflected in EEG is [25]. A similar approach to the one discussed before is presented in [25]. The data were collected from 12 subjects and features were selected using Fisher criteria and forward feature selection. A classification accuracy of 92.8% was achieved using SVM. Moreover, in [26], an approach for the classification of EEG with three different levels of valence and arousal is presented. Two different experimental setups were experimented. Features were extracted from the time and frequency domains. The best performing model utilizes a Deep Net with RELU dropout and 10-fold cross validation. The achieved accuracies in the two setups are 79% and 95.5%, respectively.

Additionally, a specific type of attention was the focus in [27]. In this study, the auditory object-specific attention in 13 subjects was analyzed using EEG. Several entropy-based features were used to build two classification models. The first one is based on linear discriminant analysis, while the second is an SVM-based model. The best obtained result was 76.9% using the SVM-based model.

Many studies in the field of cognitive-affective states detection focus on emotion recognition, which has a great overlap in the used features with the EEG features. For instance, in [28], a deep learning model is utilized to capture the spatial and frequency characteristics of the multi-channel EEG signals for the ultimate goal of modeling human emotions. More specifically, arousal, valence, and dominance were detected with accuracies of 68.2%, 66.7%, and 67.2%, respectively. A dataset called DEAP is considered in this study, which is a dataset for emotion analysis using physiological signals. In this dataset, samples were collected from 32 subjects watching 40 different videos that stimulate different emotions. In [29], a similar approach is presented; however, the accuracy was much enhanced. The model employed in this study was based on recurrent neural networks, which was able to achieve an accuracy of 90.8%, and 91% for valence and arousal classification tasks, respectively, on the DEAP dataset. This finding proves that EEG has some spatial information that needs to be considered when classifying emotions. Additionally, in [30], covariate shift adaptation is considered in the automatic detection of emotions. Many models were investigated including SVM, deep learning with principle component analysis (PCA), and deep learning with PCA and covariate shift adaptation. However, the best achieved results did not exceed 53% for the detection of arousal. In [31], another approach for the classification of human emotions using DEAP dataset was investigated. The extracted features were fused using Restricted Boltzmann machines (RBM). The best achieved accuracy was 76.8% for detecting valence.

A similar study was presented in [32], where that authors also utilized a deep learning framework for emotion recognition tested on the DEAP dataset. Their innovation was clear in the feature extraction technique as they mapped spatial characteristics, frequency domain, and temporal characteristics in two-dimensional images. Additionally, they made a significant contribution by proposing a hybrid neural network called CLRNN, which integrates Convolution Neural Networks (CNN) and Long Short-Term-Memory (LSTM) Recurrent Neural Networks (RNN). This network managed to achieve an average classification accuracy of 75.21%. Moreover, in [33], the authors investigated the possibility of measuring situational interest in classrooms. EEG signals of 43 students were decomposed using Empirical Mode Decomposition (EMD), while T-tests and Receiver Operator Characteristics (ROC) were used to rank the resulting Intrinsic Mode Functions (IMFs). For each participant, a matrix of the best six features from four EEG channels. These features were used to build classification models with both Support Vector Machine (SVM) and K-Nearest Neighbor (KNN) classifiers with 10 cross-validations. SVM outperformed KNN as the SVM model achieved an accuracy of 93.3% and 87.5% for two different data sets collected by the authors. KNN achieved a lower accuracy of 87.5% and 86.7% over the same two data sets. Another interesting finding of the study is that gamma and delta bands are the most efficient in detecting situational interest.

Since 2013, several researchers have been working on creating benchmark datasets to be publicly available for engagement recognition such as DAiSEE [34], HBCU [35], in-the-wild [36], and SDMATH [37]. For instance, Hernandez et al. [38] were the first to attempt the problem of recognizing user engagement. They modeled the problem of determining engagement of a TV viewer as a binary classification task, using facial geometric features and SVMs for classification. They created a very small custom dataset, labeled by a single coder for the presence of engagement. Additionally, Whitehill et al. [39] compared several models (Gentle AdaBoost, SVM with Gabor features, multinomial logistic regression) for automatic cognitive engagement detection using facial expression. They found that, for binary classification (e.g., high engagement versus low engagement), the accuracy of SVM with Gabor features is comparable to that of human observers (about 70%). The experiments were conducted on the HBCU dataset. Moreover, Gupta et al. [40] presented a dataset annotated for user engagement, boredom, confusion, and frustration in the wild, called DAiSEE. They developed several models (InceptionNet frame level, InceptionNet video level, convolutional three-dimensional (3D) CNN, Long-Term recurrent convolutional network) for detection. Long-Term recurrent convolutional network was the best model, obtaining an accuracy of 58% for user engagement. Kaur et al. [41] introduced an “in-the-wild” dataset, which was categorized with four engagement levels. The authors employed a multiple kernel learning (MKL) SVM model and obtained an accuracy of 50.77%. After ambiguous labels (where labels tend to disagree) were removed from the dataset, the authors improved the performance up to 75.77%.

Other studies were introduced to map facial expressions to affective states in learning environments. For instance, in [42] the authors examined facial characteristics to detect the affective states (emotions) that accompany the conceptual material’s deep-level learning. Videos of the faces of the participants were recorded while interacting on computer literacy subjects with AutoTutor. The student’s affective states (boredom, confusion, pleasure, flow, frustration, and surprise) were recognized after the tutoring session. After the tutoring session, the learner, a peer, and two trained judges recognized the student’s affective states (boredom, confusion, pleasure, flow, frustration, and surprise). Two autonomous judges coded the facial expressions of the participants using the Facial Action Coding System (FACS) of Ekman. Correlational analysis showed that particular facial characteristics could segregate confusion, pleasure, and frustration in the baseline state of neutrality, but boredom was indistinguishable from neutral. Based on extremely diagnostic facial features, the authors discuss the prospects of automatically identifying these feelings. The participants are 28 undergraduate students whose faces were recorded and later shown to judges that were instructed to make judgments on what affective states were present in each 20-s interval. Inter-judge reliability was computed using Cohen’s kappa for all possible pairs of judges. Additionally, the study in [43] attempts to detect three levels of Cognitive Load (CL) in participants taking an arithmetic test under the presence and absence of affective interference (e.g., arousal and other emotional and psychological factors). For example, it is human nature to have a mixture of high cognitive load and personal feelings (e.g., frustration, confusion) when introduced to complicated tasks. The goal of the study is to collect EEG signal and facial videos from subjects undergoing complicated tasks, to ultimately model the cognitive states and skills under emotional stimulus.

## 3. Materials and Methods

### 3.1. Data Acquisition

#### 3.1.1. Electroencephalogram (EEG) Recording

EEG signals were recorded using the 14-channel wireless headset (Emotiv Epoc, San Francisco, CA, USA). Besides the 14 channels recording the signals, the headset has two channels that work as reference points. The 14 channels are distributed around different parts of the scalp based on the 10–20 system. Figure 1 shows the distribution of sensors’ locations over the scalp. This figure is particularly important as the names and locations of the channels are critical to the discussion of the modalities of cognitive states and skills. The sampling frequency utilized during recording is 128 Hz. In addition to recording, the headset also works as a brain computer interface (BCI) that generates performance metrics every 10 s. These performance metrics scores include: excitement, frustration, engagement, relaxation, focus, and stress. These performance metrics are used and validated in different applications: feedback during exams, rehabilitation, and sports [44]. The engagement score is especially of interest, as it will be used to label the collected EEG signals as engaged or non-engaged in building the engagement model.

#### 3.1.2. Facial Expressions Recording

To collect as much facial features (FF) as possible and make sure that the device can be cost-efficient, we deployed a webcam with a resolution of 640 × 480 Pixels. We sat the webcam to directly face the subjects and focused directly on their faces during recording while they were performing the tests. The videos were recorded in AVI format since it is virtually compatible with all devices. Then, the videos were converted into frames in which the FF are mapped in order to facilitate the processing of the videos. The exact details of the mapping of the facial landmarks and the extraction of the FF will be discussed later in the feature extraction section.

### 3.2. Experimental Setup

#### 3.2.1. Target Population

In this study all university students that met the inclusion criteria and were available for the actual experiments were equally targeted. Neurological disorders were among the exclusion criteria, including any dementias, such as epilepsy and Alzheimer disease, and all cerebrovascular diseases, such as stroke and migraine, in addition to traumatic disorders and brain tumors. Other exclusion criteria included skin allergy and any medication that may interfere with the physiological data. The subjects signed a consent informing them about the experiment, its purpose, procedures, the discomfort they might be subjected to while using the headset and the incentive they would receive as an appreciation to their time and contribution.

As a result, 127 subjects (64 males, 63 females) with ages ranging from 18 to 26 years (mean = 20.7 years, standard deviation = 1.49 years) were included in this study. The following figures show some demographics about the included population. The data collection was extended across 6 weeks, where five subjects, on average, were included each day. Each session lasted for an average of 55 min, including the instructions given to the participants and the setup time. Experiments were held in a room with internet connectivity and good lighting. Figure 2 shows the experimental design of one subject taking the test next to an observer, while Figure 3 shows a closer look at the screen of the observer measuring both the EEG signals and the video recording the facial expressions simultaneously.

#### 3.2.2. Procedure and Task Description

The project aimed at analyzing the two modalities EEG and facial expressions associated with different cognitive states and skills. The models were divided into three categories: engagement models, instantaneous attention models, and complex cognitive skills models. The engagement models and the instantaneous attention models utilized the data collected during a continuous performance task provided by the Psychology Experiment Building Language (PEBL) test battery. On the other hand, the complex cognitive skills models utilized the data collected during the general cognitive assessment provided by the Cognifit test battery.

(1) Continuous Performance Task

While wearing the headset and recording the video, the subject takes the PEBL continuous performance task (PCPT). In PCPT, participants are exposed to a series of letters appearing on the screen for 14 min. The letters, 1 inch in size, appear one by one in white font on a black screen. The participant’s task is to respond to the letters by pressing space bar, except for the letter X, which they should not respond to. After the end of the test, a complete report of the responses of the subject is generated [45]. This record is in the format of binary scores mapped to each trial depending on whether the participant was successful in hitting the spacebar. In addition, a score for sustained attention is added based on the participant’s hit response time and rates of omissions and commissions. This report will be used in labelling the data in case of the instantaneous attention model. In case of the engagement, the labelling will be based on the engagement score generated by Emotiv BCI. The reason for choosing this test specifically was to ensure that the subjects naturally show engaged and unengaged states without the introduction of complex tasks. Moreover, the test requires a minimal amount of movements to ensure the quality of the videos recorded, and hereby, the quality of the facial expression measured.

(2) General Cognitive Assessment

The subjects were subjected to cognitive tests [46] while wearing the headset and during video recording. This cognitive test is clinically approved for measuring cognitive skills. The test measures 23 cognitive skills through 16 tests. The paper focuses on the three cognitive skills of interest to the educational applications: focused attention, planning, and shifting. The focused attention skill measures the ability to detect relevant stimuli, which is very essential while studying and solving problems. As for planning, it measures the ability to anticipate the future to reach a certain goal, which is critical to solving problems and keeping track of lectures. Shifting measures cognitive flexibility, which is an integral part of learning and capturing information from the environment. These cognitive skills were scored between 0 and 800, where a score < 200 was categorized as low, a score of 200–400 was moderate, and a score ≥ 400 was high.

### 3.3. EEG Artifact Identification and Removal

One of the biggest drawbacks of EEG is its high susceptibility to be coupled with many sources of artifacts. These artifacts mainly include distortions from eye, muscles, pulse, respiratory, and sweat. The effect of these artifacts on the quality of analysis has been proved in either clinical or practical experiments. Hence, many manual and automatic methodologies have been developed to remove artifacts from EEG without distortion of the original neural signals. The first step of decomposition commonly utilizes an Independent Component Analysis (ICA), which assumes the signal being linear or linear mixtures of brain sources and artifacts. The second step involves the removal of the components labeled as artifacts and then reconstructing the signals [47]. EEGLAB Matlab toolbox (version 2019.0) is used to apply ICA with a runica algorithm, which is the recommended algorithm based on the EEGLAB documentations [48].

There are two approaches to label the artifacts after running ICA. First, manual inspection of the resultant component activations, maps, and spectra. This approach has many drawbacks. First, to be able to do this task better than the second approach, it needs extensive training and expertise. Second, introducing a subject factor to our experiment will make it impossible to replicate. Third, it is highly time-wasting considering the size of the dataset we are considering. Hence, we decided to follow the second approach, the automatic labeling. EEGlab provides several extensions to deal with this task specifically; one of their most commonly used extensions is ICLabel, a seven-category IC classifier using a neural network trained on hundreds of thousands of ICs. Moreover, it is not a binary classification, as it classifies components of the brain, eye, muscle, line nose, channel nose, and others. Furthermore, it allows changing the range of probability to mark components as an artifact for each of these categories independently, which allows a higher range of tweaking the algorithm [49]. Different combinations of threshold probability were used. The one with the best fit to the collected data was rejecting components labeled as eye, muscle, line nose, channel nose, or others with probability higher than 0.8.

The data collected from the 127 subjects were saved in 171 EEG records as some participants preferred a break between the two cognitive tests. Out of the 171 records, 29 records have no components removed, 36 have one component removed, 40 records have two components removed, 38 have three components removed, 15 records have four components removed, and 13 records have five components removed or more. Among the affected channels, the ones closer to the movement of facial muscles are more likely to be removed in the denoising procedure. AF3 and AF4 are the two channels with the highest frequency of component removal (about 50 and 35 components were removed from each one). In addition, FC6 is the least affected channel. There is also a trend that the channels from measuring the left side of the brain are more likely to be removed with respect to the corresponding channels on the right side.

The signal to noise ratio (SNR) of the entire dataset was calculated across the 14 channels. The summation of the 14 channels SNR scores is 301.84 decibels (dB). The median of the SNR scores per channel is 21.24 dB, with a maximum value per channel of 25.32 dB and a minimum SNR score per channel of 19.91 dB (with a standard deviation of 1.36).

### 3.4. Feature Extraction

#### 3.4.1. EEG Feature Extraction

(1) Time-Domain Features

In the signal’s time domain, the extracted features are primarily statistical ones. The extracted time-domain features include the minimum, maximum, mean, variance, standard deviation, coefficient of variance, kurtosis, skewness, 1st quartile, 2nd quartile, 3rd quartile, Shapiro–Wilk test statistic, Shapiro–Wilk test p-value, Hjorth mobility, and Complexity parameters.

Each of the above features is calculated for each of the 14 channels. In addition, the Hjorth mobility and complexity parameters are calculated as given by Equations (1) and (2) [50], respectively:(1)$$Mobility=\sqrt{\frac{var\left(\frac{dx\left(t\right)}{dt}\right)}{var\left(x\left(t\right)\right)}}$$
(2)$$Complexity=\frac{Mobility\left(\frac{dx\left(t\right)}{dt}\right)}{Mobility\left(x\left(t\right)\right)}$$
where x(t) is the signal, var is its variance, and the forward difference formula is used for the calculation of $\frac{dx\left(t\right)}{dt}$.

Thus, in total, there are 210 time-domain features extracted for each participant.

(2) Frequency Domain Features

1 Wavelet Transform

EEG oscillations are described by frequency, power, and phase. Changes in the rhythmic activity is correlated to tasks demands such as perception and cognition. Transforming the collected EEG from time domain to frequency domain is an important step as there are many features not shown in the time domain. The cognitive activity of the brain can be presented as the power of the four main frequency bands; theta (4–8 Hz), alpha (8–12 Hz), beta (16–32 Hz), and gamma (32–63 Hz), as suggested in cognitive physiology. As a result, the EEG signals features were calculated from calculating the relative power of the four main frequency sub-bands in the frequency domain in addition to calculating the average power in each channel. EEG are nonstationary, time-varying signals; thus, to extract a frequency band specific information from it, a convolution process is used. Convolution is a mathematical process that relies on dot product, which is particularly useful for time-frequency decomposition because this interpretation will facilitate understanding how to extract power and phase angles from complex numbers, where one vector is considered the signal while the other is considered the kernel. This convolution is done through what is called a wavelet. Its main function is to isolate frequency-band-specific activity and to localize it in time to decompose the signal for feature extraction. Its function is to show the level of extent the EEG features are similar to the applied wavelet [51].

A Morlet wavelet looks like a sine wave in the middle but then tapers off to zero at both ends. A Morlet wavelet is a gaussian window that has no sharp edges that produces artifacts, dampens the influence of surrounding time points on the estimate of frequency characteristics at each time point, and allows you to control the trade-off between temporal precision and frequency precision [52]. To make a Morlet wavelet, a sine wave is created then creates a Gaussian having the same number of time points and the same sampling rate, then multiplying them point by point. Wavelet convolution uses many wavelets of different frequencies, their numbers are not constrained and can be of varied frequencies. Their frequencies are the same as those of the sine wave and are called a peak frequency [53,54].

The discrete signal f(n) is filtered more than on time up to a certain predetermined level. This filtration contains two filter pass filters; a low pass filter for getting the approximation coefficient (CA) and high pass filter for getting the detailed coefficient (CD) [55]. Daubechies 8 (db8) wavelet is used as the mother wavelet, where 8 represents the number of vanishing moments. It is used as a wavelet function, since it is known to be one of the most efficient functions to be used with EEG signals. The EEG signals have an interest range of 0–50 Hz, while the input signal is collected with a band of 0–500 Hz. Hence, it is necessary to decompose the signal to the 8th level in order to fully obtain the bands with the lowest frequency. Considering Table 1, which shows the relationship between the wavelet components and frequencies, the first three components are labeled as noise because they are out of the range of interest. The EEG powers and relative powers for different sub bands frequencies were calculated from the wavelet coefficients. The relative powers can be calculated by dividing the power of each sub-band to the total power of the signal. The relative powers will be used later for classification [53,54]. The energy of the delta sub-band is given by (3), while that of the theta, alpha, and beta sub-bands are given by (4):(3)$$E\left[A_{6}\right]=\overset{N}{\underset{j=1}{\sum }}{\left|A_{6}\right|}^{2}$$

Energy of:(4)$$E\left[D_{l}\right]=\overset{N}{\underset{j=1}{\sum }}{\left|D_{l}\right|}^{2}\;\;\;\;\;\;\;\;\;\;\;l=6,5,4$$

As j = 6, the total energy of EEG epoch is given by (5):(5)$$E_{total}=E\left[A_{6}\right]+\overset{6}{\underset{j=1}{\sum }}\left(E\left[D_{j}\right]\right)\;E_{delta}=\frac{E\left[A_{6}\right]}{E_{total}}\;E_{theta/alpha/beta}=\frac{E\left[D_{i}\right]}{E_{total}}$$
where *i* = 6,5,4 for beta, alpha, and beta, respectively.

2 Baseline Normalization

The presence of individual differences between the subjects such as hair and skull thickness make it difficult to aggregate the raw power values across the subjects; this is solved by using baseline normalization. Decibel (dB) conversion, a widely used method for baseline normalizations in cognitive electrophysiology, was used in this paper, where the normalized power was calculated through the change in the activity power in relation to the baseline power as shown in Equation (6) [51]:(6)$$Normalized\;Power=10\cdot log_{10}\left(\frac{activity\;power_{\;tf}}{baseline\;power{\;}_{f}}\right)$$
where t and f indicate time and frequency points, respectively.

Before starting the experiment, the baseline power was measured for each subject through recording their EEG signals while resting with open eyes for 15 s and with closed eyes for another 15 s, in addition to a 6 s window during which the subject got ready for the experiment. This normalization was also used to remove any unrelated factors that interfere with the results.

#### 3.4.2. Facial Expressions Feature Extraction

(1) Facial Expressions Feature Extraction for engagement models

Dlib library [56] was used to determine the location of the 68 (x,y) coordinates of the facial landmarks (see Figure 4) as raw features that were used to extract new features to build the classifiers, as described below.

1 Eye aspect ratio (EAR)

Based on the work by Soukupová and Čech in their 2016 study [57], Real-Time Eye Blink Detection using Facial Landmarks, the eye aspect ratio for the right eye can be defined as the ratio of the height to width as follows:(7)$$EAR=\frac{\left\|p_{42}-p_{38}\|+\|p_{41}-p_{39}\right\|}{2\|p_{40}-p_{37}\|}$$
where ‖‖ is the Euclidean distance.

2 Mouth aspect ratio (MAR)

Similar to EAR, the mouth aspect ratio can be defined as:(8)$$MAR=\frac{\left\|p_{68}-p_{62}\|+\|p_{67}-p_{63}\|+\|p_{66}-p_{64}\right\|}{3\|p_{55}-p_{49}\|}$$

3 Nose to Jaw (NTJ)

To detect the left/right pose of a learner, the normalized distance between the nose tip and the right jaw is defined as:(9)$$NTJ=\frac{\left\|p_{31}-p_{3}\right\|}{\left\|p_{15}-p_{3}\right\|}$$

4 Noise to Chin (NTC)

To detect the up/down pose of a learner, the normalized distance between the nose tip and the chin is defined as:(10)$$NTC=\frac{2\|p_{31}-p_{9}\|}{\left\|p_{22}-p_{8}\|+\|p_{23}-p_{10}\right\|}$$

Besides the facial landmarks features approach, an automatic feature extraction technique was applied where the face was directly fed to the Convolutional Neural Network (CNN). The model utilizes the early layers to play the role of feature extraction. Different architectures for CNN with various parameters were explored, and the best model obtained is shown in Figure 5.

(2) Facial Expressions Feature Extraction for instantaneous models and cognitive skills models

Since there is no clear mapping between what the models are predicting and the known facial landmarks, another approach was implemented for the feature extraction for the remaining analyzed states other than engagement. The approach is based on combined deep and shallow learning techniques. The features of the recorded video frames while participants are performing the instantaneous attention test and the cognitive assessment battery were extracted using the pre-trained deep network GoogleNet [58]. GoogleNet is a deep convolutional network with nine inception modules, four convolutional layers, four max-pooling layers, three average pooling layers, five fully-connected layers, and three softmax layers. The reason for choosing this model is that GoogleNet, as compared to other pretrained models (such as AlexNet or ResNet, etc.), has bottleneck layers and a global pooling layer, which reduces the number of parameters [59], and hence, has enhanced real-time performance.

As preprocessing steps, the videos collected during a given time window were sequenced into frames. Then, the images were centered and resized to 224 × 224, which is the input size of the GoogleNet input layer. The sequence of frames was then applied to GoogleNet and the activations of the last pooling layer are extracted as the feature vector. The size of this feature vector is 1024. As a final step, the feature vector for the video was the average of the extracted feature vector averaged across the frames. Finally, the extracted feature vector was used to train a shallow learning classifier.

### 3.5. Cognition Detection Shallow Models

Several machine learning shallow models were tested for the analyzed cognitive states using either EEG or facial features. In this section, the theoretical foundations of the best performing classification models are laid out.

#### 3.5.1. Random Forest (RF)

In this model, several separate decision trees at the training phase are constructed. To make a final prediction, the constructed trees are pooled; in particular, the mode of the classes is computed. To decrease node impurity, feature importance (the higher the value, the more important the feature is) is calculated weighted by the probability of reaching that node. This probability is calculated by the number of samples that reach the node, divided by the total number of samples. If we assume only two child nodes are allowed, node importance is calculated by:
$ni_{j}=w_{j}c_{j}-w_{left\left(j\right)}c_{left\left(j\right)}-w_{right\left(j\right)}c_{right\left(j\right)}$, where$ni_{j}$ is the importance of node j$w_{j}$ is the weighted number of samples reaching node j$c_{j}$ is the impurity value of node jleft(j) and right(j) are the child nodes from left and right split on node j, respectively.

The importance of feature i (denoted by $\;f_{i}$) in a decision tree is calculated as:(11)$$f_{i}=\frac{\sum_{ni\;splits\;on\;feature\;j}(ni_{j})}{\sum_{k\in Set\;of\;all\;nodes}(ni_{k})}$$

The feature’s importance is then normalized as follows:(12)$$normalized\;f_{i}=\frac{f_{i}}{\sum_{j\in Set\;of\;all\;features}f_{j}}$$

The final feature importance that RF outputs (denoted by $RFf_{i}$) is the average over all trees:(13)$$RFf_{i}=\frac{\sum_{j\in set\;of\;all\;trees\;}\left(normalized\;f_{ij}\right)}{T},\;$$
where $normalized\;f_{ij}$ is the normalized feature importance for node i in tree j and T is the total number of trees [60].

#### 3.5.2. Logistic Regression (LR)

LR is a predictive analysis model that can be used in classification problems. It transforms its output using the sigmoid function to return a probability value, which can then be mapped to two or more discrete classes. The sigmoid function is used to map predicted values to probabilities (between 0 and 1) as follows:(14)$$h_{\theta }\left(X\right)=\frac{1}{1+e^{\beta_{0}+\beta_{1}X}}$$

In order to transform this to a discrete class, a threshold τ is selected (e.g., τ=0.5), which will classify values into either class 1 or class 2 as follows:
if p≥ τ, then output class 1if p<τ, then output class 2
where p is the probability returned by LR that the observation being positive (or belonging to class 1).

The cost function to be minimized is [60]:(15)$$J\left(\theta \right)=\frac{-1}{m}\sum_{i=1}^{m}\left[y^{\left(i\right)}{}^{}\log\left(h_{\theta }\left(x^{\left(i\right)}\right)\right)+\left(1-y^{\left(i\right)}\right)\log\left(1-h_{\theta }\left(x^{\left(i\right)}\right)\right)\right],\;$$
where m is the number of classes.

#### 3.5.3. Linear Support Vector Machines (Linear SVM)

Linear Support Vector Machines, or LSVM, is one of the methods that is used for classification and regression analysis. Simply it is about finding a linear hyperplane that separates the different categories from each other. To do so, it defines the best linear hyperplane that has a maximum distance from the nearest elements from either category. First, support vectors are set that each touches the boundary of the categories. They are defined by:(16)$$H_{1}=wx_{i}+b=+1\;H_{2}=wx_{i}+b=-1$$
where H_1_ and H_2_ represent class 1 and 2, respectively, and w is a weight vector, x is an input vector, and b  is the bias. $H_{0}$ is the separator line between both classes and equals the median of $H_{1}$ and $H_{2}$ (i.e., $H_{0}=wx_{i}+b=0$) Then, an optimization algorithm is deployed to find the weights that maximizes the distance between the support vectors [60].

#### 3.5.4. Extra-tree Classifier (ET)

This classifier is also called “Extremely Randomized Trees Classifier,” which is an ensemble learning technique. This output of this technique is a combination of a forest consisting of all the results of multiple de-correlated decision trees making it similar to Random Forest Classifier (RF). The only difference between ET and RF is its manner in constructing the decision tree. In ET forest each decision tree is built from the original training sample then each one of them is randomized with a k features a random sample at each node. These features are selected from the feature set where each set the best fit features enabling the splitting of the data; mainly based on Gini Index. This method of randomization of the features creates multiple de-correlated decision trees [60].

#### 3.5.5. Multilayer Perceptron (MLP)

MLP classifiers consist of an input layer, an output layer, and one or more hidden layers in between. Each node i in layer j  is connected with every other node in the following layer with a weight $w_{ij}$. The weighted inputs are summed and passed through an activation function. Two of the common activation functions are sigmoid $a\left(x\right)={\left(1+x\right)}^{-1}$ (which ranges from 0 to 1) and the hyperbolic tangent tanh(x), which ranges from −1 to 1.

The supervised learning process is performed by changing connection weights $w_{ij}’s$ after each data point is processed, based on the amount of error in the output compared to the true label. This is done through backpropagation. The degree of error in an output node j in the nth training example can be calculated by $\left(n\right)=y_{j}\left(n\right)-d_{j}\left(n\right)$, where $d_{j}$ is the output produced and $y_{j}$ is the true label. The weights can then be adjusted based on the corrections that minimize the output as follows. $E\left(n\right)=\frac{1}{2}\sum_{j}^{}e^{2}\left(n\right)$. The error is then propagated back through the network, one layer at a time, and the weights are updated according to the amount that they contributed to the error. One of the classical training algorithms that carries out this job is the stochastic gradient descent [61].

## 4. Analysis and Discussion

### 4.1. Collected Data

As discussed in the experimental setup section, EEG signals and video recordings were collected from 127 subjects while taking PCPT test and the general cognitive assessment. However, the data from 18 subjects have been rejected, leaving the data from 109 subjects, which will be used in building the model. The causes of rejections include: abnormal EEG signals, withdrawal of the subject in the middle of the recording session, internet shortage during the cognitive assessment, subject experiencing itching of the scalp during the recording session, the EEG headset running out of battery during the recording, and other technical difficulties. Each of the 109 subjects has 17 EEG records (one for PCPT test and 16 for the cognitive tests) and 17 video recordings.

### 4.2. Analysis of the Cognition Scores

#### 4.2.1. Engagement Models

The engagement models are trained and tested on 10 s windows and on 60 s windows. As discussed before, the engagement model depends on the data collected during the PCPT test, whose duration is 14 min. In other words, the data collected 109 subjects are actually 9156 windows in the case of 10 s models, and 1526 windows in case of 60 s models. It should be mentioned that the scientifically validated test batteries that were used did not include a measure for engagement; hence, the engagement scores generated by the headset software were employed. The labelling of windows is based on the engagement scores generated by Emotiv BCI every 10 s. The threshold of labelling the windows as engaged or non-engaged is selected to be the median of all the scores, which is 0.56493. Figure 6 and Figure 7 show the distribution of the two classes between all the 10 s and 60 s windows.

#### 4.2.2. Instantaneous Attention Models

The binary labels for the instantaneous attention were evaluated based on the number of mistakes each participant has conducted during the 60-sec and 10-sec time windows as mentioned before. Figure 8 and Figure 9 show how the labels vary across the two time steps. As shown in the figures, the data are imbalanced with the minority class of low attention. Moreover, the imbalance ratio is larger in the case of a 60-sec time window as compared to that of the 10-sec time window.

#### 4.2.3. Cognitive Skills Models

The cognitive test battery reports the scores of all of the 23 skills (includes the three skills we chose to analyze) out of 800. All the subjects were divided into three classes for each of the three skills. Subject with score < 200 is categorized as low, 200 ≤ score < 400 as moderate and score ≥ 400 as high. This is the same categorization scheme recommended by the test designers [36]. Figure 10, Figure 11 and Figure 12 shows the distribution of the three classes in the three selected skills. As shown in the figures, the class with the low score has the largest number of samples for all the skills.

### 4.3. Evaluation Criterion

All the models have been developed for educational purposes to create an interactive feedback mechanism between the teacher or the earning management system (LMS). The ultimate goal of this mechanism is to introduce a certain change when the learner is at the low state of cognition. For instance, if the engagement model detected the learner at a non-engaged state, it should notify the teacher or the LMS to introduce a stimulus to recapture the learner’s engagement. Consequently, the detection of the low state has a higher weight than other classes in this type of application. This motivated the utilization of the $F_{2}$ score as the main evaluation criterion while comparing the resultant models. $F_{2}$ is a variation of the F-beta score that gives the recall higher weight than precision. The formula for the $F_{2}$ score is as follows:(17)$$F_{2}\;score=\left(1+2^{2}\right)\cdot \frac{Precision*Recall}{\left(2^{2}\cdot Precision\right)+Recall}$$

In order for this formula to fit our purpose, the low class has to be set as the positive class in all of the models, which is the case for all of the reported results. Besides the $F_{2}$ score, precision-recall curves are also used in case of models with close $F_{2}$ scores in order to ensure the favor of the model of the best consistent trade-off between precision and recall. It is worth mentioning that the data were split into 80% for the training phase, while the 20% is kept for the evaluation and reporting of the performance measures in case of engagement and instantaneous attention models. In case of the cognitive skills, the train test split was on 70–30% basis. This is mainly to be able to reach a reliable $F_{2}$ score on the test set since the sample size is small. It is worth mentioning that, for all models, the data were shuffled in each iteration of the training process.

### 4.4. Obtained Results and Discussions

#### 4.4.1. Classification Results Using EEG-Based Models

(1) Classification of Engagement

The labels of engagement are obtained based on a scaled engagement score (with range from 0 to 1) generated by the Emotiv BCI, which generates a score every 10 s. Two types of models are examined: 10 s models and 60 s models. First, the 10 s models utilize a direct mapping between a 10 s EEG signal record and the engagement score generated during that interval. Second, the 60 s models utilize a mapping between 60 s EEG signal record and the median of the six engagement scores generated during that interval. However, the 60-sec window features were extracted directly from the 60-sec signal to avoid the correlation between the 10-sec window features; hence, to be able to build a different set of models that consider how different cues about the analyzed states might appear in longer time windows. Different standardization and normalization techniques of the features have been tested. Standard scaling of the features leads to the best performance. The dataset was divided into training and testing with 80% and 20% ratios, respectively. The training dataset was resampled on a 10-fold cross-validation basis. Then, the performance of the models was reported based on the prediction of the models on the testing dataset. Five machine learning classification models from Python Scikit-learn library (www.scikit-learn.org) was tested: Random Forests (RF), Multi-layer Perceptron (MLP), k-Nearest Neighbors (KNN), Support Vector Classifier (SVC), and Quadratic Discriminant Analysis (QDA). Hyper parameter tuning of all the models was applied searching for the best F2 score.

The models utilizing the frequency-domain features only resulted in a similar or better performance than the models utilizing both the time-domain and frequency-domain features. This suggests a dependence of engagement detection on the power features only. Moreover, the models examining 60-sec windows consistently performed better than the models examining 10-sec windows. For a time window of 60 s, the best performing model is RF (with Gini criteria, number of estimators = 1000). The $F_{2}$ score obtained is 0.861 (cross-validation score is 0.881 +/− 0.01), with almost the same accuracy score of 0.862. RF was also the best performing model in the case of a 10 s window. In that case, the $F_{2}$ score obtained was 0.715 (cross-validation score is 0.75 +/− 0.04) and the accuracy score was 0.735. Figure 13 and Figure 14 show the $F_{2}$ scores for the built models for 60-sec and 10-sec time windows, respectively. Despite the very close $F_{2}$ scores obtained from the five models, the precision-recall curves of the models show RF consistently achieving better trade-off between precision and recall than the other models as shown in Figure 15 and Figure 16.

(2) Classification of Instantaneous Attention

The EEG signals of the participants while taking the PCPT test were divided into time windows each of 60 sec and 10 sec, respectively. The test lasted for 14 min on average; thus, a total of 14 and 84 samples per participant were analyzed for the 60 sec and 10 sec time splits, respectively. The labels for each sample were generated from the output of the PCPT, where the number of mistakes made by the participant in each time window of the test was used to indicate their attention level and was mapped to 0 or 1 to indicate low or high instantaneous attention, respectively. The rule used is as follows. Participants with less than m mistakes during the given time window were considered as highly attentive and those with m or more mistakes were considered to be in a low attention state, where m is a threshold that was determined by experiments to be equal to 4. This threshold value was chosen based on the analysis of the dependence between engagement and attention scores, where a value of 4 produced a high dependence measure between the two scores. To study the relation between the engagement and instantaneous attention scores, the Chi-square test for independence was applied. The results obtained show that the two scores were dependent at the level of significance of 0.05 with X-squared = 4.9619, df = 1, and *p*-value = 0.02591.

It is worth mentioning that the collected samples are imbalanced using the two time windows. For instance, a total of 202 and 1158 samples for low and high attention, respectively, in the 60-sec time split is obtained. Moreover, the samples contain outliers across most of the features. Hence, as a preprocessing step, robust scaling is applied to deal with outliers, in which the data is centered based on the interquartile range, in addition to balancing the training data using synthetic minority oversampling (SMOTE) [62]. This oversampling approach yielded approximately equal samples for the two classes (953 for class 0 and 922 for class 1).

For a time window of 60 s, the best classification results for the attention were obtained using RF classifier (with Gini criteria, number of estimators = 100). $F_{2}$ score obtained is equal to 0.82 (cross-validation score is 0.83 +/− 0.02) with test accuracy of 0.866. Similarly, for a time window of 10 s, the $F_{2}$ score obtained is equal to 0.72 (cross-validation score is 0.75 +/− 0.03) with a test accuracy of 0.55. Figure 17 and Figure 18 show the $F_{2}$ scores for the built models for 60-sec and 10-sec time windows, respectively. As shown in the figures, the RF classifier gives the best f-2 scores in the two cases. The low performance in the 10-sec time window as compared to the 60-sec one might be due to the fact that the features related to attention in the EEG signal need more time to be reflected in the time and frequency domains. Moreover, Figure 19 and Figure 20 show the precision vs. recall curve for the best performing models using the 60-sec and 10-sec time windows, respectively. As shown in the figures, the average precision for the 10-sec time windows is much lower that of the 60-sec ones. It should be mentioned that the baseline $F_{2}$ score for the 60-sec time window model (using the highly-imbalanced data, as shown in Figure 8) is 0.46.

(3) Classification of Complex Cognitive Skills

As mentioned earlier, 23 complex cognitive skills were measured by the general cognitive assessment test. Out of the 23 skills assessed by the test, four skills have been selected to be investigated based on their relevance to educational applications. The dataset consisting of 109 samples was divided into three classes (low, moderate, and high) based on the scores of each skill. A grid search has been performed on the best four models and the $F_{2}$ scores were calculated using the best-performing parameters for each classifier. Before training, all the extracted features undergo a preprocessing procedure, starting with standard scalar transformation, which rescale the features to have a standard decoration of 1 and a mean of 0. Next, feature importance values are calculated using a logistic regression model, such that the higher the relevance of the feature to the modality, the higher its importance score. Figure 21, Figure 22 and Figure 23 show the $F_{2}$ scores of the four skills evaluated on the testing dataset.

The top model for the case of FA skill is the MLP, with a single hidden layer of 50 nodes and a logistic activation scoring 0.63 F2 score. Regarding the Planning skill, the highest scoring model was Extra Trees (with 400 estimators) scoring, with an F2 score of 0.68. Finally, for the Shifting skill Extra Trees (with 100 estimators) also gave the highest F2 score among all the classifiers, with 0.68 F2 score. The parameters for all the models used were set using grid search. It should be mentioned that the baseline F2 scores for Focused Attention, Planning, and Shifting skills models are 0.16, 0.14, and 0.12, respectively.

#### 4.4.2. Classification Results Using Facial Expression-Based Models

(1) Classification of Engagement

The approach utilized in these models is a frame-based approach, where the model is basically used to individually classify whether a learner is engaged or not in each frame extracted from the video sequence, and then, an aggregate module is utilized to obtain the overall engagement class for the video sequence. Hence, using the labels generated by Emotiv BCI to train these models did not fit as a single score is generated every 10 s by BCI. In other words, 300 frames will be given the same label regardless of the differences between them, which does not fit with a frame-based approach. Instead, 30,000 frames from different video sequences have been extracted from the DAiSEE dataset, which is a crowd annotated dataset [34]. Then, the frames were split into training/validation/test with the following percentages, respectively: 60%, 20%, and 20%. There are 23,409 images (about 78%) annotated as engaged and the rest are non-engaged based on the pose angle and eye blinking. Linear support Vector Classifier (Lin-SVC), RBF kernel Support Vector Classifier (RBF-SVC), and Random Forest (RF) were examined with the facial landmarks features extracted. Moreover, a CNN model, which automatically extracts features, was also examined. The details of the facial landmarks features and the architecture of the CNN were discussed in the feature extraction section. The three shallow models performed very badly, with RF as the best model scoring only an F2 score of 0.402. However, the CNN outperformed them significantly scoring an F2 score of 0.824 and an accuracy score of 0.925. Figure 24 shows the model accuracy across training and validation epochs.

In order to utilize this frame-based approach to predict engagement, this procedure was applied on a 1614 video sequence of 10 seconds collected from the dataset collected by the team. The class predicted by the CNN model was compared with the engagement class generated by Emotiv BCI. The same procedure was applied again on 267 video sequences of 60 s. The results of this evaluation gave a high consensus of engaged cases (82%, 84%). Non-engaged cases gave a lower consensus of (54%, 60%) for the 10 s and 60 s windows, respectively, while the EEG model performed better in non-engaged cases. This high disparity between the non-engaged cases was expected, as a non-engaged mental state does not necessarily lead to physical changes in facial expression that can be detected by crowd annotation. This suggests the superiority of annotation with BCIs over human manual annotation in case of applications that gives more weight for non-engaged cases detection.

(2) Classification of Instantaneous Attention

Similar to what was done in analyzing the EEG, the facial expressions of the participants while taking the PCPT test are divided into clips each of 60 and 10 s and annotated with the binary labels that were generated when annotating the EEG samples. For a time window of 60 s, the best classification results for the attention using RF classifier (with Gini criteria, number of estimators = 1000) are as follows. The $F_{2}$ score obtained is equal to 0.81 (cross-validation score is 0.82 +/− 0.02) with a test accuracy of 0.74. Similarly, for a time window of 10 s, $F_{2}$ score obtained is equal to 0.79 (cross-validation score is 0.82 +/− 0.01) with a test accuracy of 0.79. Figure 25 and Figure 26 show the $F_{2}$ scores for the built models for 60-sec and 10-sec time windows using the videos, respectively. As shown in the figures, the RF classifier gives the best $F_{2}$ scores in the two cases. Moreover, Figure 27 and Figure 28 show the precision vs. recall curve for the best performing models using the 60-sec and 10-sec clips of attention videos, respectively. As shown in the figures, the average precision for the 10-sec clips is much lower that of the 60-sec ones.

Thus, it is noted that the performance of facial expression-based models for the instantaneous attention is approximately equal to that of the EEG. One possible reason is that attention can be easily detected from facial expressions. In addition, for video-based models and based on the test accuracy, a 10-sec model has a slightly better performance in comparison with the 60-sec one; this is due to the fact that the number of samples for a 10-sec model is larger than that of the 60-sec model. Hence, the performance of GoogleNet as a feature extraction step is enhanced.

(3) Classification of the General Cognitive Skills

The same preprocessing applied on the EEG features in case of the complex cognitive skills discussed previously was applied also to the video features. Figure 29, Figure 30 and Figure 31 show the $F_{2}$ scores of the four skills evaluated on the testing dataset.

Classification of the FA skill was more difficult using video features. The top model in this case is the MLP, with a single hidden layer of 50 nodes and a logistic activation. These parameters were obtained by performing a grid search. Regarding the Planning skill, the highest scoring model was the Extra Trees (with 100 estimators) scoring, with an $F_{2}$ score of 0.78. Finally, for the Shifting skill, Extra Trees (with 400 estimators) also gave the highest $F_{2}$ score among all the classifiers, namely, 0.81. The parameters of all the models used were set using grid search.

It is worth mentioning that some experiments were performed to assess the importance of the features used. For the engagement, for instance, the following sets of features were experimented: the time-domain features only, the frequency domain features only, and then the combined features. The performance of the model that employs the time domain features only was quite poor and the performances of the models that employ the frequency domain only and the combined features were almost the same. Hence, all of the frequency features (70 features) were employed given the good number of samples (9156 for the 10 s models and 1526 for the 60 s models). Common feature selection techniques were tested, and they all led to a drop in the performances of the models, similar to the instantaneous attention models. Hence, all of the frequency domain features calculated from the signals of the 14 electrodes seem to hold information about engagement. This hypothesis was further asserted by the fact that the evaluation of the features’ importance values obtained from the best-performing model has a very low standard deviation (~0.003, for the engagement). For the cognitive skills, such as planning, perception, and working memory, since the sample size is smaller than that used for the other cognitive states (engagement and instantaneous attention), the threat of overfitting is more likely to happen. Hence, feature selection methods were experimented as part of the modeling process. However, these methods did not contribute to the accuracy and F2 scores, and at best, they did not lower them. The best feature selection method is based on the Extra Trees classifier. In addition, for these skills, the most important features were the statistical features. Moreover, to test that the models did not overfit, 20% of the used dataset were removed and kept for the testing phase. For all of the analyzed targets, the differences between the performance of the models on the training and the testing phase were not dramatically decreased (which usually occurs as a sign of overfitting), with that of the testing phase being decreased, naturally.

Other experiments were performed to assess features importance to the built models. This was investigated using the values of features importance with the best-performing EEG model for each of the five cognitive skills and states. This was not investigated for the video models, because their features were extracted using GoogleNet; hence, the actual meaning of the extracted features is unknown. Figure 32 shows the top five features and their importance scores for the engagement model. As shown in the figure, the right hemisphere is more influential than the left hemisphere, since the top 10 features in terms of feature importance values contain nine features calculated from the signals of channels in the right hemisphere. Moreover, Channel T8 is specifically influential, as four of the top 10 features are power features calculated from the signal of this channel.

For instantaneous attention, Figure 33 shows the top five features’ importance for the best performing model. As shown in the figure, the features with the top importance to the model are mainly from the frequency domain, and they belong to the parietal, frontal, and occipital lobes, which are responsible for attention, thinking, and problem solving.

For cognitive skills such as planning, perception, and working memory, the time domain features are more influential, as shown in Figure 34, Figure 35 and Figure 36.

Table 2 summarizes the results of the best performing models using the two modalities. For both the engagement and instantaneous attention models, the best EEG based model performed slightly better than the best facial expressions-based model. On the other hand, the best facial expressions-based model performed notably better than the best EEG based model for both planning and shifting. As for focused attention, both modalities failed to achieve good performance, with the best EEG based model performing relatively better. The reason for this may be due to the difficulty of differentiating the levels of focused attention from a short time window, since in a previous study [19], EEG-based models for detecting focused attention were implemented with accuracy of 0.85. However, the window of analysis was relatively long (about 30 min long); hence, it was not suitable for a real-time system.

## 5. Conclusions and Future Work

In this paper, an approach for the automatic prediction of some cognitive states is presented. Two cognitive states, engagement and instantaneous attention, and three cognitive skills, focused attention, planning, and shifting, have been investigated using both EEG and facial expressions modalities.

Apart from assessing the performance, the analysis performed in the paper led to interesting conclusions about the nature of the modalities examined. First, the models based on a 60 s time window consistently performed better than the models based on a 10 s time window in case of the EEG based models by a big margin. However, the two time windows do not show a significant effect on the performance of the facial expressions-based models, as the performance was almost the same. It is worth mentioning that the performance of EEG based engagement models was not affected by the exclusion of the time domain features, which suggests the complete dependence of the model on the frequency domain features. In addition, the comparison between the predictions of engagement models based on the Emotiv BCI labels and engagement models based on human labeling of video frames showed a high consensus in the engaged classes. However, the consensus was low for the non-engaged classes, with the EEG based model achieving better performance. This suggests that using human annotation of data (for example crowdsourcing) is not a good option compared with annotation based on BCIs for applications that are more interested in detecting the non-engaged cases.

The data were collected in an experimental setting; hence, as a future work, further experimentation needs to be done in a real classroom or an online-learning setting to test the effectiveness of the proposed approach in a real life system. Moreover, combining the two modalities of the EEG and facial expression holds potential to achieve better performance. Currently, this point is under investigation. In addition, other cognitive skills can also be modeled to build complete cognitive profiles about learners in an automatic way.

## Figures and Tables

**Figure 1 sensors-20-03516-f001:**
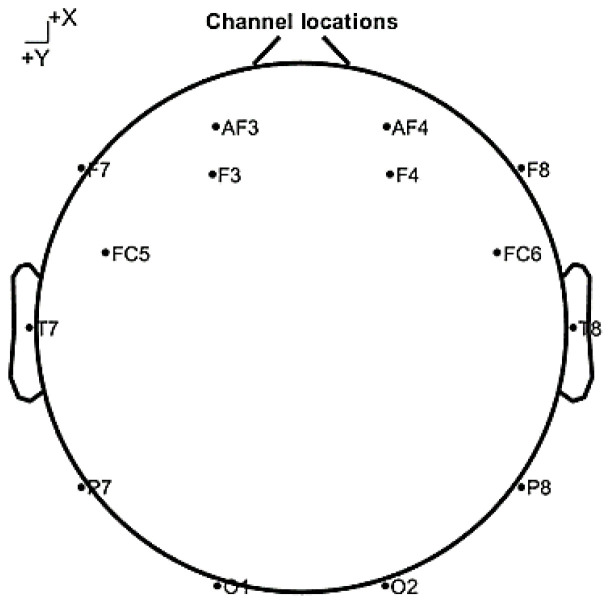
A view of the distribution of the sensors’ locations that was utilized by the 14 channels headset.

**Figure 2 sensors-20-03516-f002:**
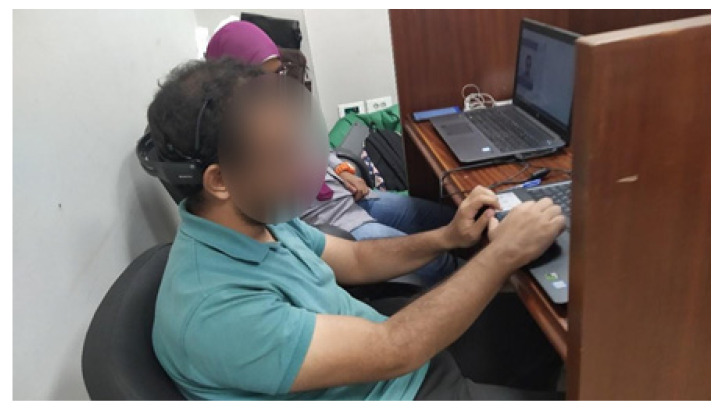
The experimental setup showing a subject undergoing the experiment next to an observer.

**Figure 3 sensors-20-03516-f003:**
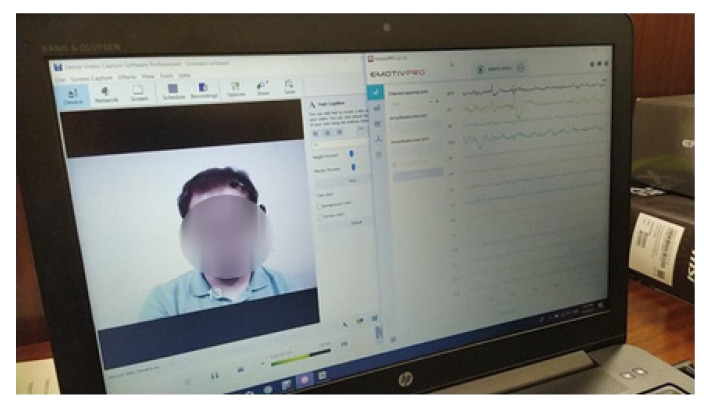
Video and EEG recording from the observer screen.

**Figure 4 sensors-20-03516-f004:**
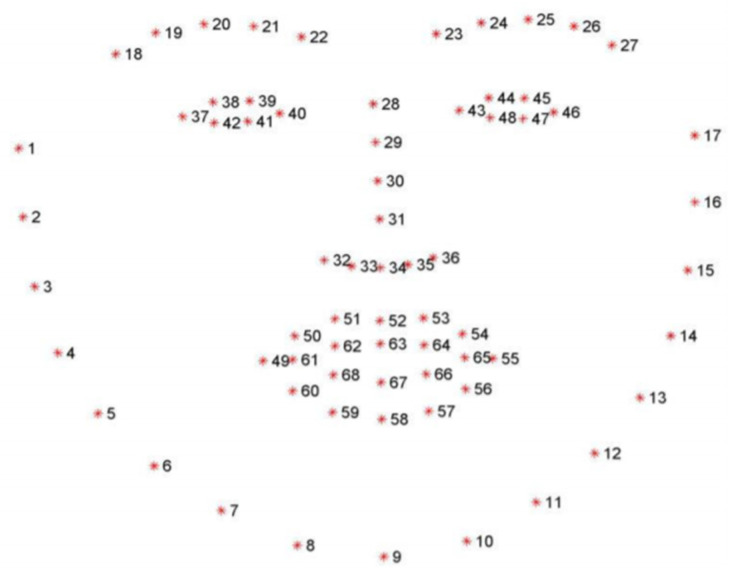
Dlib facial landmarks coordinates that were used to calculate the features of the video-based classification for engagement.

**Figure 5 sensors-20-03516-f005:**
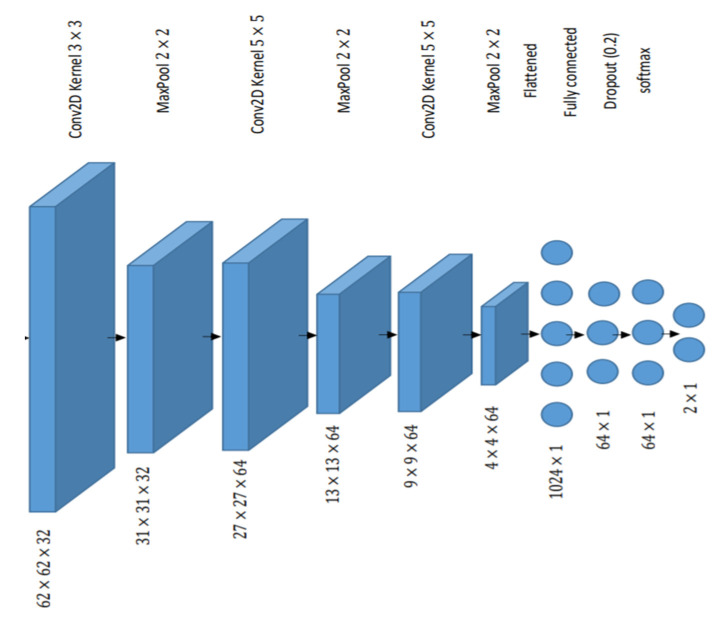
Architecture of the Convolution Neural Networks (CNN) model that was used in video-based classification of engagement.

**Figure 6 sensors-20-03516-f006:**
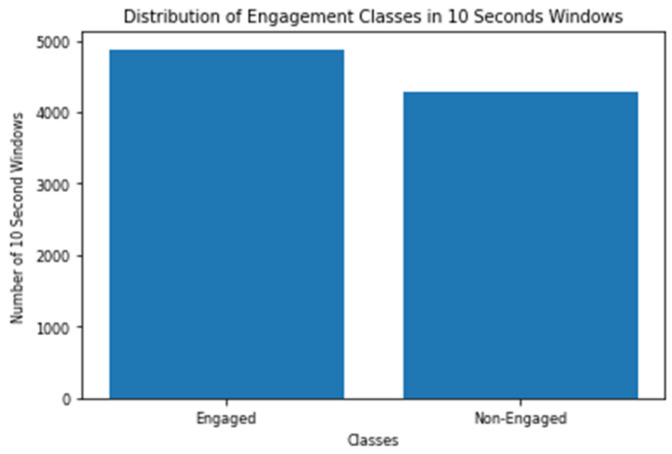
Distribution of Engagement Classes in 10 Seconds Windows.

**Figure 7 sensors-20-03516-f007:**
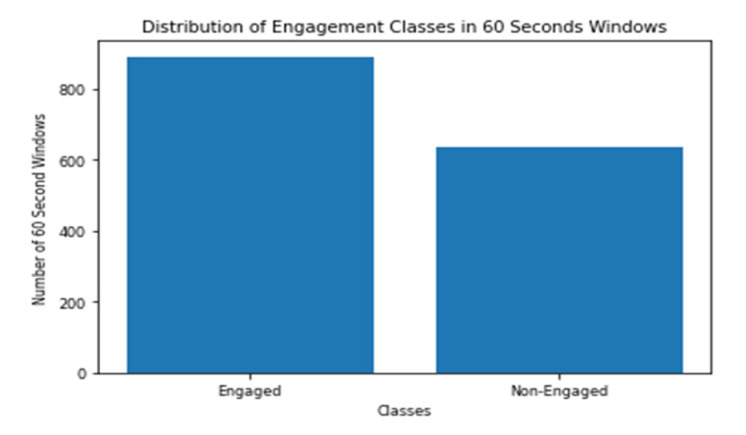
Distribution of Engagement Classes in 60 Seconds Windows.

**Figure 8 sensors-20-03516-f008:**
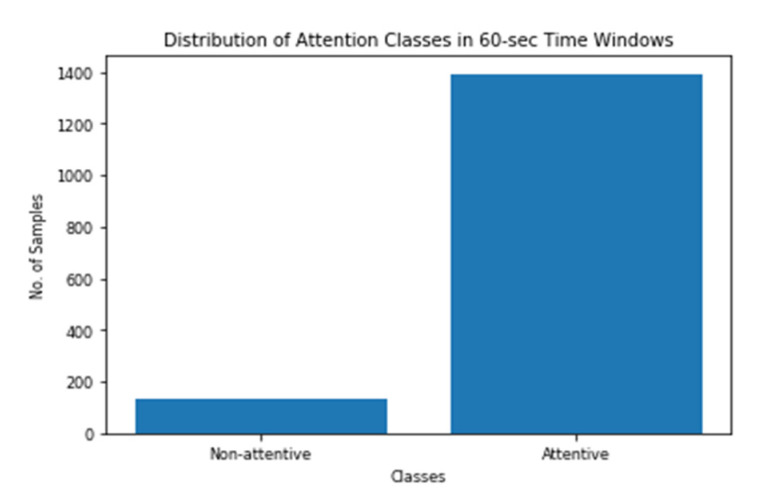
Distribution of Attention Classes in 60-sec Time Windows.

**Figure 9 sensors-20-03516-f009:**
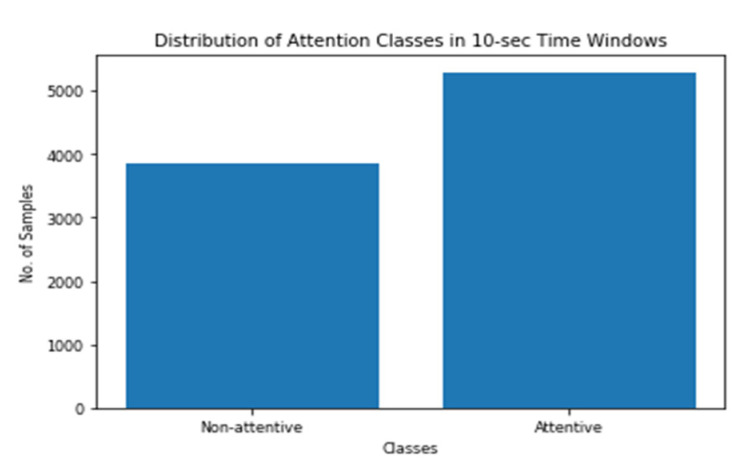
Distribution of Attention Classes in 10-sec Time Windows.

**Figure 10 sensors-20-03516-f010:**
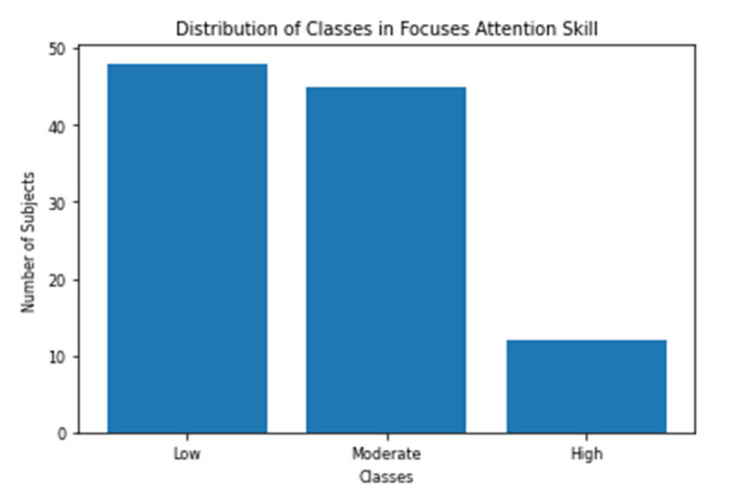
Distribution of Focused Attention (FA) classes.

**Figure 11 sensors-20-03516-f011:**
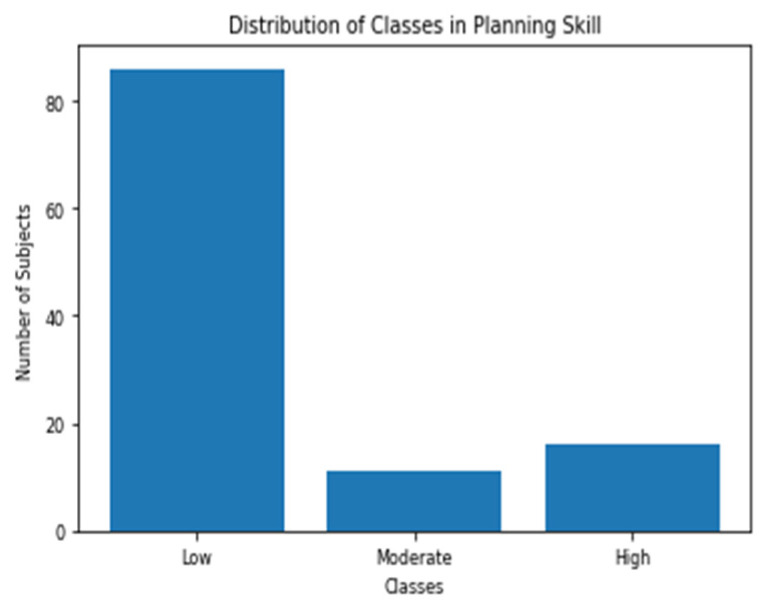
Distribution of Planning classes.

**Figure 12 sensors-20-03516-f012:**
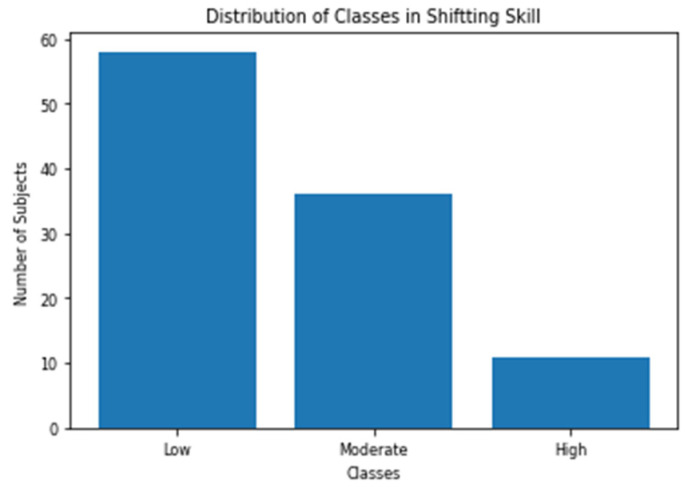
Distribution of Shifting classes.

**Figure 13 sensors-20-03516-f013:**
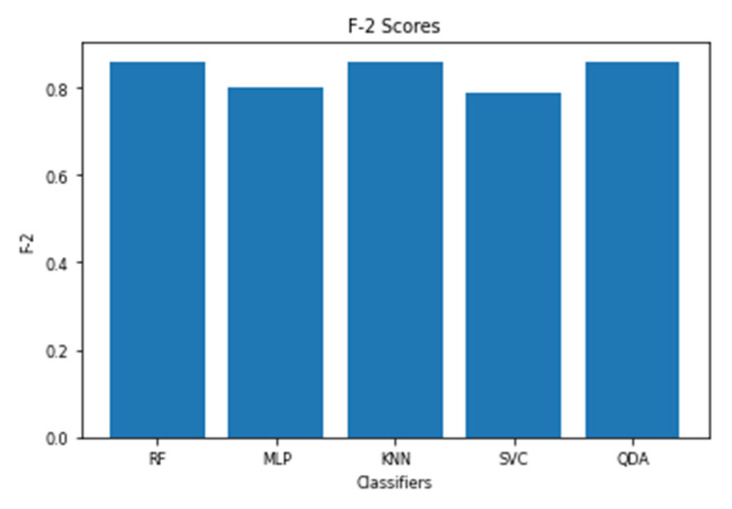
The $F_{2}$ scores for the engagement models using EEG a 60-sec time window.

**Figure 14 sensors-20-03516-f014:**
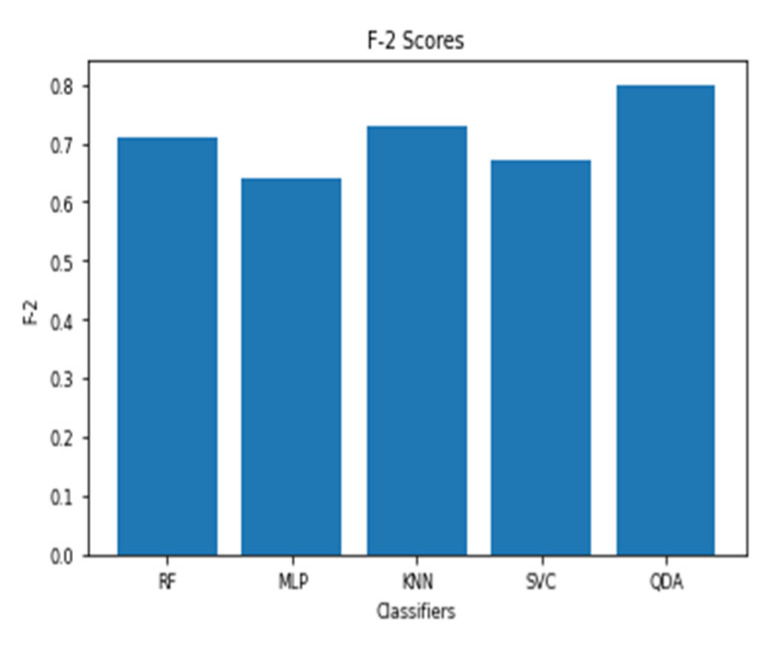
The $F_{2}$ scores for the engagement models using EEG a 10-sec time window.

**Figure 15 sensors-20-03516-f015:**
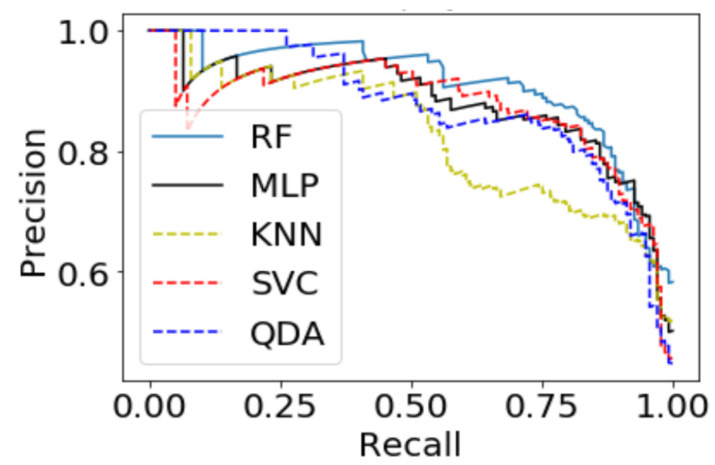
Precision-Recall curves of the 60-s engagement models after hyper-parameter tuning.

**Figure 16 sensors-20-03516-f016:**
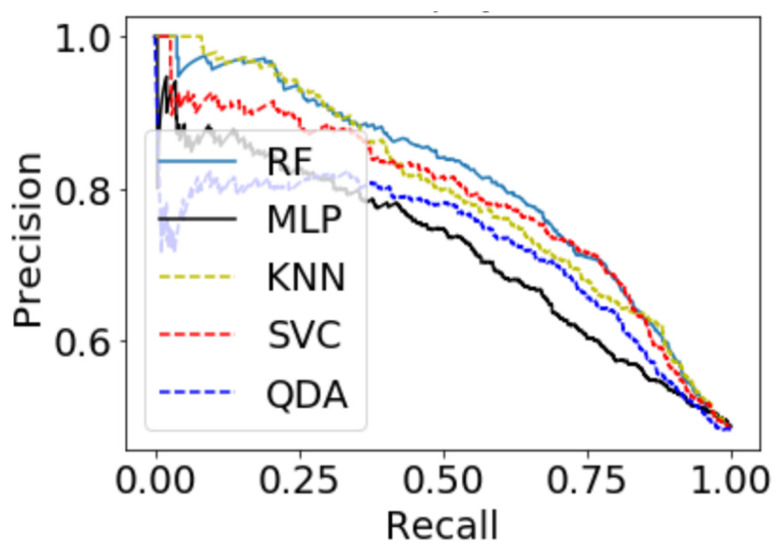
Precision-Recall curves of the 10-s engagement models after hyper-parameter tuning.

**Figure 17 sensors-20-03516-f017:**
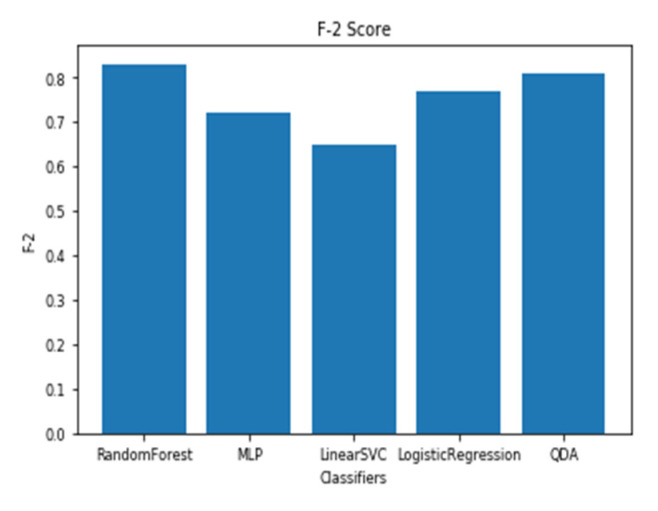
The $F_{2}$ scores for instantaneous attention models using EEG with a 60-sec time window.

**Figure 18 sensors-20-03516-f018:**
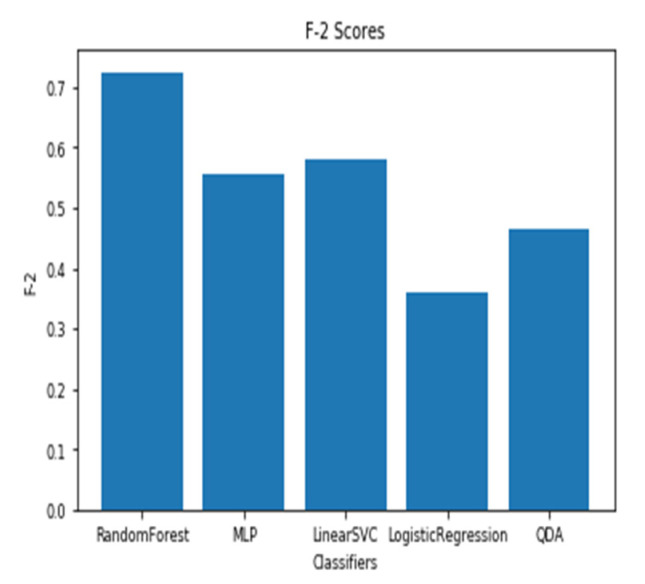
The $F_{2}$ scores for instantaneous attention models using EEG with a 10-sec time window.

**Figure 19 sensors-20-03516-f019:**
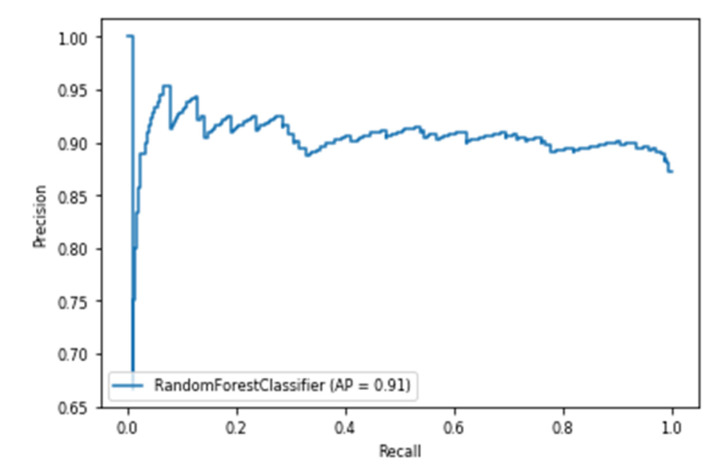
The precision vs. recall curve for the best-performing EEG-based model for attention using 60-sec time windows.

**Figure 20 sensors-20-03516-f020:**
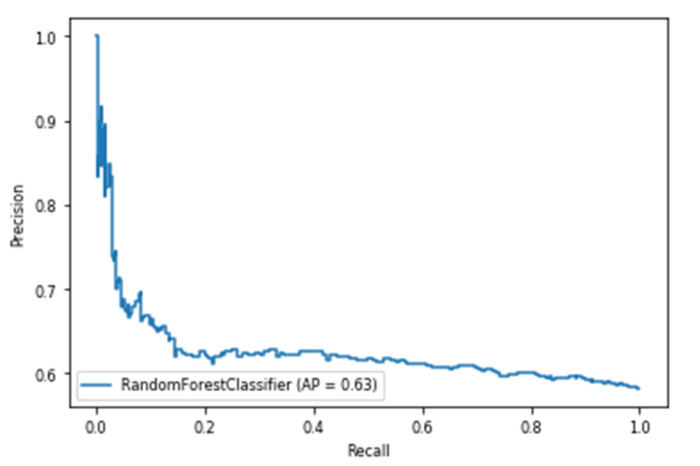
The precision vs. recall curve for the best-performing EEG-based model for attention using 10-sec time windows.

**Figure 21 sensors-20-03516-f021:**
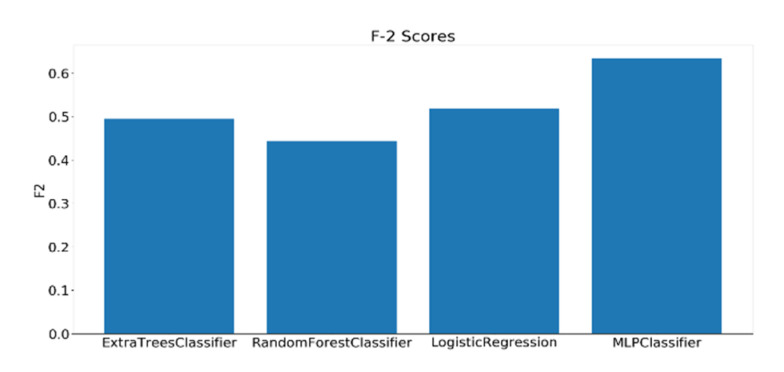
$F_{2}$ scores for the classification of Focused Attention using EEG features.

**Figure 22 sensors-20-03516-f022:**
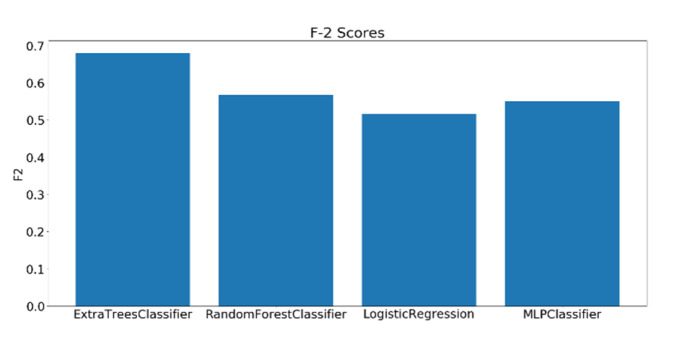
$F_{2}$ scores for the classification of Planning using EEG features.

**Figure 23 sensors-20-03516-f023:**
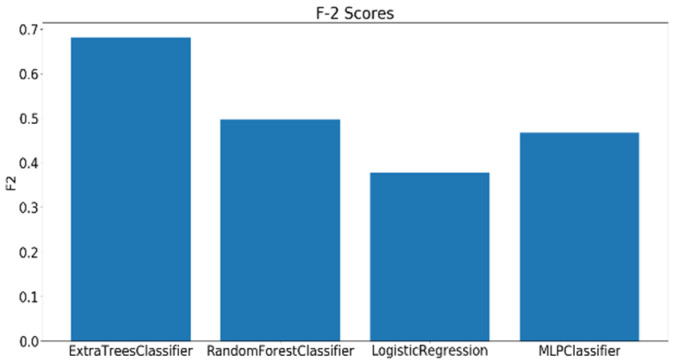
$F_{2}$ scores for the classification of Shifting using EEG features.

**Figure 24 sensors-20-03516-f024:**
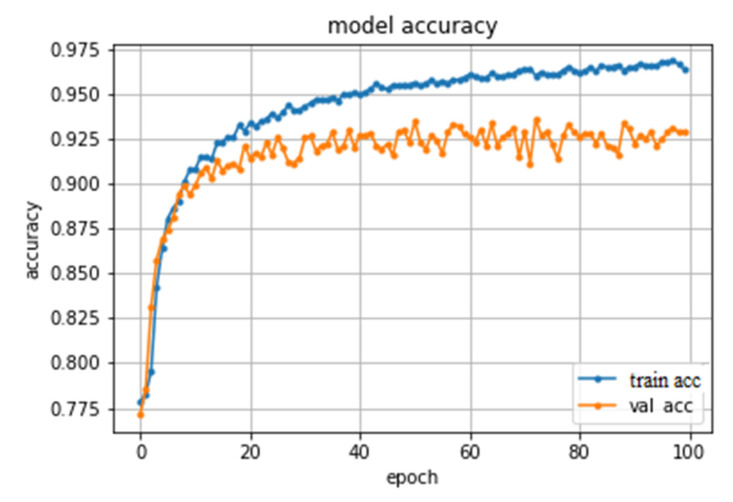
The CNN model accuracy across training and validation epochs.

**Figure 25 sensors-20-03516-f025:**
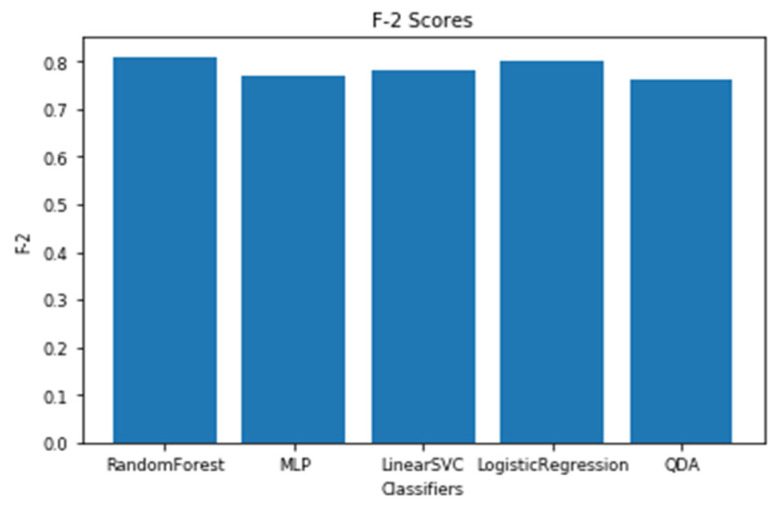
$F_{2}$ scores for instantaneous attention models using video features for a 60-sec time window.

**Figure 26 sensors-20-03516-f026:**
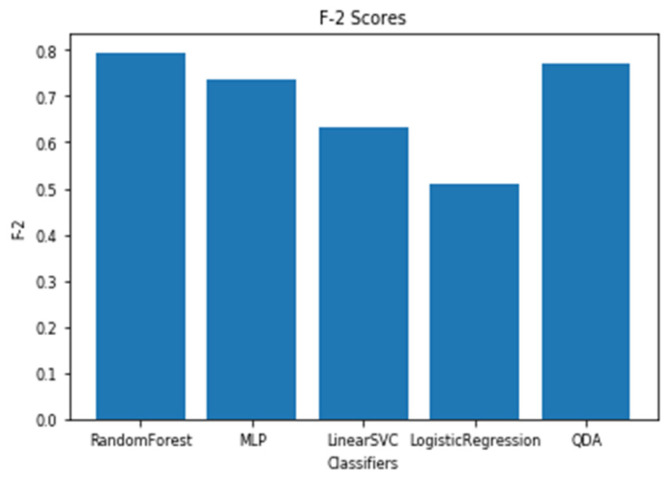
$F_{2}$ scores for instantaneous attention models using video features for a 10-sec time window.

**Figure 27 sensors-20-03516-f027:**
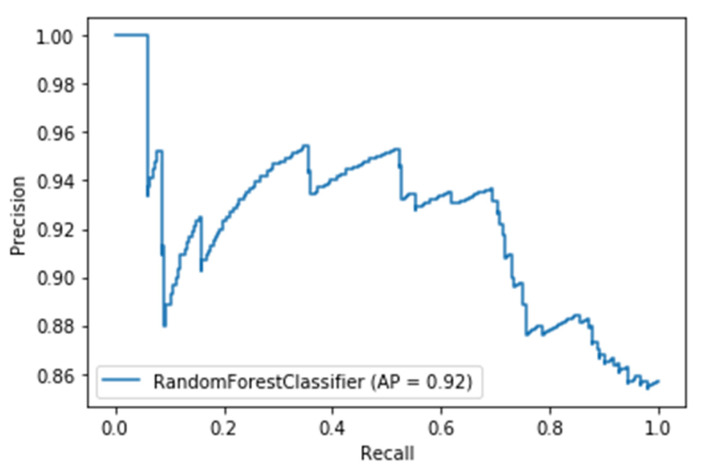
The precision vs. recall curve for the best-performing video-based model for attention using 60-sec clips.

**Figure 28 sensors-20-03516-f028:**
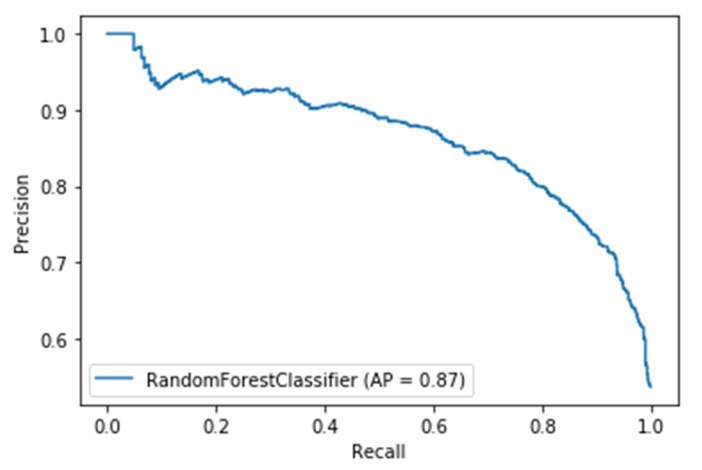
The precision vs. recall curve for the best-performing video-based model for attention using 10-sec clips.

**Figure 29 sensors-20-03516-f029:**
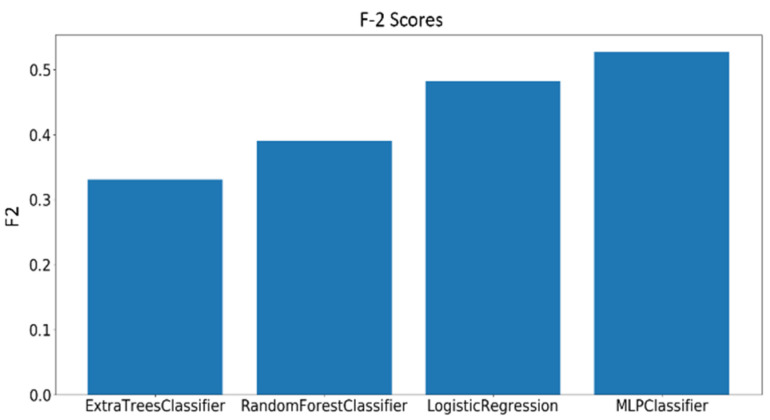
$F_{2}$ scores for the classification of Focused Attention using video features.

**Figure 30 sensors-20-03516-f030:**
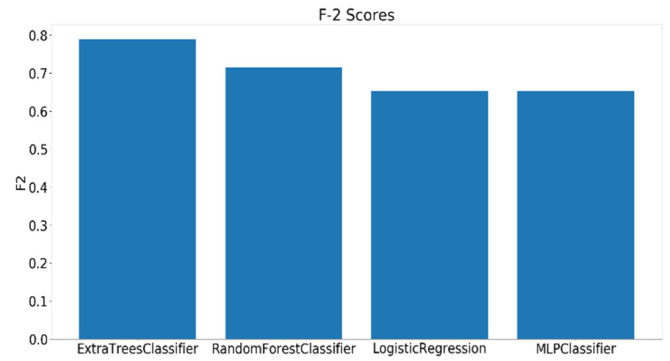
$F_{2}$ scores for the classification of Planning using video features.

**Figure 31 sensors-20-03516-f031:**
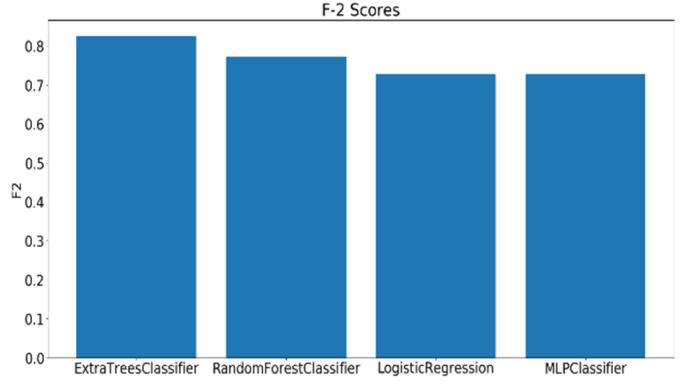
$F_{2}$ scores for the classification of Shifting using video features.

**Figure 32 sensors-20-03516-f032:**
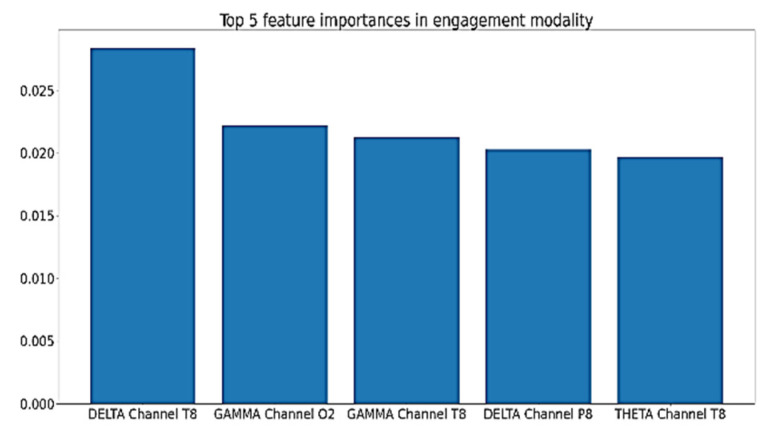
Top five features importance in the best performing engagement model.

**Figure 33 sensors-20-03516-f033:**
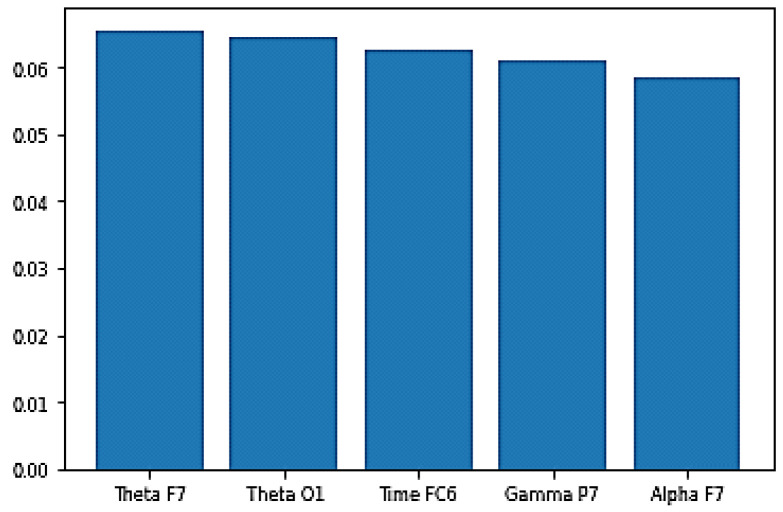
Top five features importance in the best performing instantaneous attention model.

**Figure 34 sensors-20-03516-f034:**
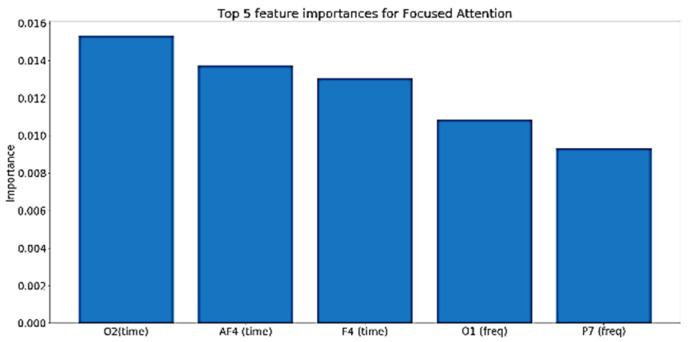
Top five features importance in the best performing Focused Attention model.

**Figure 35 sensors-20-03516-f035:**
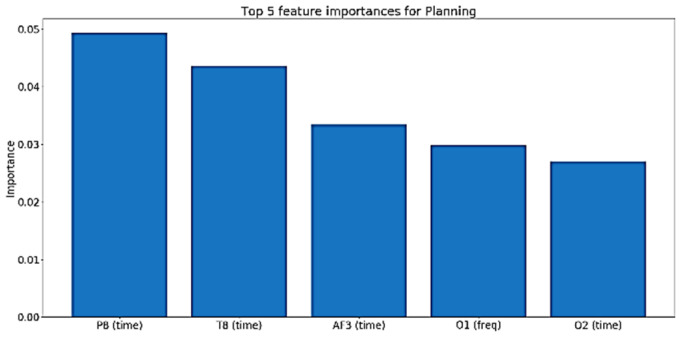
Top five features importance in the best performing Planning model.

**Figure 36 sensors-20-03516-f036:**
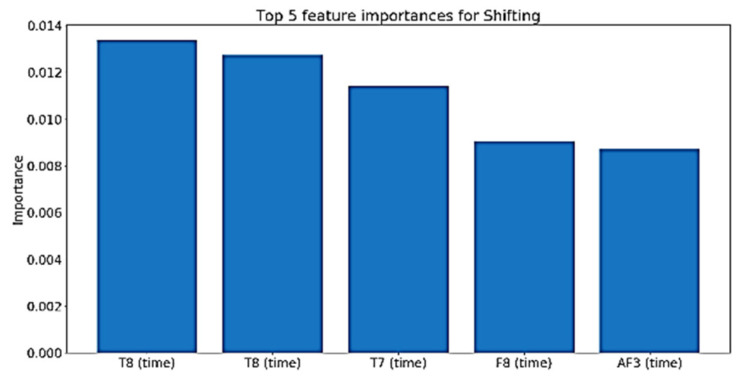
Top five features importance in the best performing Shifting model.

**Table 1 sensors-20-03516-t001:** The relationship between Wavelet coefficients and frequencies and signal information.

Wavelet Coefficient	Frequency (Hz)	Signal Information
D1	250–500	Noise
D2	125–250	Noise
D3	63–125	Noise
D4	32–63	Gamma
D4	16–32	Beta
D6	8–16	Alpha
D7	4–6	Theta
D8	0–4	Delta

**Table 2 sensors-20-03516-t002:** A summary of the best-performing models.

Focus	Best Model	F2 Score
EEG-based Engagement	Random Forests (with Gini criteria, number of estimators = 1000)	0.86
Facial expression-based Engagement	Convolutional Neural Network (See Figure 5 for the architecture)	0.82
EEG-based Instantaneous Attention	Random Forests (with Gini criteria, number of estimators = 100)	0.82
Facial expression-based Instantaneous Attention	Random Forests (with Gini criteria, number of estimators = 1000)	0.81
EEG-based Focused Attention	Multilayer Perceptron (a single layer with 50 nodes and a logistic activation)	0.63
Facial expression-based Focused Attention	Multilayer Perceptron	0.53
EEG-based Planning	Extra Trees (with Gini criteria and number of estimators = 400)	0.68
Facial expression-based Planning	Extra Trees	0.78
EEG-based Shifting	Extra Trees (with Gini criteria and number of estimators = 100)	0.68
Facial expression-based Shifting	Extra Trees	0.81

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
