# Peer review of "The Automatic Detection of Cognition Using EEG and Facial Expressions"

_sensors, 2020, doi:10.3390/s20123516_

Round 1

Reviewer 1 Report

El Kerdawy et al. in their paper nicely shows a very interesting application of EEG and facial expressions in detecting cognitive states and skills of healthy subjects. The sample size is good and the methodological design is innovative. I have only one major concern:

  • EEG analysis: It is unclear to me if the authors have used all the time-frequency domain features from each channel in their classification models. The use of an exaggerated numbers of features carries the risk of overfitting the classification performance of their model despite cross-validation. I recommend performing a feature selection strategy (i.e. by removing redundant information of each feature f in this classifier as the decrease in classification performance when the values of f are randomly permuted for each combination of channels) in order to improve the robustness of their analysis. Moreover, no information is given about which electrodes are more influent in the accuracy of their models, this is important since different cortical regions may have more impact in cognitive skills evaluation (i.e. occipital leads for visual tasks, frontal leads for attention and working memory).

Minor comments:

  • It would be interesting to know the actual accuracy of both techniques combined (EEG and facial expressions detection) in predicting engagement.

Reviewer 2 Report

The paper is interesting, and my main concerns are on the use of the Engagement score of the Emotiv to label the data, the way in which the train-test split was performed, and some questions about the time features.

(1) In general I am skeptic of the use of the Emotiv for proper research science. However, I understand that for the application of this resarch, some form of easy-to-setup commercial headset is required. Considering that the authors are using well established tests to measure engagement, cognition, etc., I wonder why the output of the black-box Engagement score from the Emotiv instead of the results of the test. Was some form of validation/comparison between the two performed?

Moreover, what we know about the Emotiv's black box is that it uses time-frequency features to predict engagement, so it sounds to me like you're overfitting by using a dataset with the output of the engagement box to predict the same output.

Lastly, since the output of the emotiv is a number between 0-1, regression seems to be more appropriate than classification for this task.

(2) The authors state "The dataset was divided into training and testing with 80% and 20% ratios, respectively. The training dataset was resampled on a 10-fold cross-validation basis."

Since there is no explict statement as to whether shuffling was applied in the splits (and I believe by default sklearn does shuffle the dataset), I am worried here that the windows in the train/validation might be too similar. I don't think there was overlap in the windows, but still, two consecutive windows will be very correlated, so there can be leakage between the training and validation set that is inflating the results. This should be addressed or properly described.

(3) There doesn't seem to be a justification for the time-domain features extracted, which, to the best of my knowledge, are not well-established as features for detecting cognition. Some of them might be intuitive to use, but I wonder why the authors deemed the Shapiro-Wilk test statistic and p-value appropriate (also, how was this used?).

Some other comments:

- Are the 60-second windows formed by concatenation of features form the 10-second windows or are the features extracted directly from 60-second windows? Were other window sizes (e.g., 30-seconds) considered?

- Please list the hyperparameters used in the search for the best algorithms.

- For the imbalanced datasets, please report random F2 score as a baseline. When reporting results on the cross-validation set, a measure of dispersion around the mean should be provided. In figures, this can be done as a boxplot for each of the methods. In the text, as a +/- standard deviation.

- Figures 17, 18, 21-23, 25-26, 29-31 occupy too much space and could be easily replaced by a table reporting these numbers for minimum disruption while reading.

- I think the dataset collected can be of use to the wider community, and encourage the authors to consider making it publicly available.

Reviewer 3 Report

I have only one major concern: This is about the classification and the training/testing division. Was the 80-20 division (line 589 and followings) random? If yes does the testing set includes unseen participants? or has the classifier "seen" all the participants expressions? In case the training set includes all the participants and a sample of their expressions can authors test a "real world" scenario where the systems is presented with unseen participants?

Other concerns:

1) Please extend the section "Classification and engagement". After reading this section I got a bit confused, was the engagement tested separately?  Or prior of the other assessments the "non engaged samples" have been removed. Can you add a table specifying the sample sizes for all the cases tested?  

2) What was the basic SNR for the entire dataset? I can see that some pre-processing is been performed to remove artifacts and noise. Can a table or a better discussion i.e. new dataset size be added to section 3.3? Would be quite interesting to discuss the effect of the noise on the classification. I can see that a certain criteria was used to remove EEG affected channels, was the dataset being made uniform in size i.e. was the a certain channel always erased because being affected by noise for the majority of the participants?

Round 2

Reviewer 1 Report

The authors have succesfully answered all my concerns.

Reviewer 3 Report

authors have addressed my concerns